# Functional Relevance of Extracellular Vesicle-Derived Long Non-Coding and Circular RNAs in Cancer Angiogenesis

**DOI:** 10.3390/ncrna10010012

**Published:** 2024-02-06

**Authors:** José A. Peña-Flores, Daniela Muela-Campos, Rebeca Guzmán-Medrano, Diego Enríquez-Espinoza, Karla González-Alvarado

**Affiliations:** Doctoral Program in Biomedical and Stomatological Sciences, Faculty of Dentistry, Autonomous University of Chihuahua, Chihuahua 31000, Mexico; dmuela@uach.mx (D.M.-C.); rguzman@uach.mx (R.G.-M.); denriquez@uach.mx (D.E.-E.); kalvarado@uach.mx (K.G.-A.)

**Keywords:** exosomes, lncRNAs, long non-coding RNAs, circRNAs, cancer, angiogenesis, neovascularization

## Abstract

Extracellular vesicles (EVs) are defined as subcellular structures limited by a bilayer lipid membrane that function as important intercellular communication by transporting active biomolecules, such as proteins, amino acids, metabolites, and nucleic acids, including long non-coding RNAs (lncRNAs). These cargos can effectively be delivered to target cells and induce a highly variable response. LncRNAs are functional RNAs composed of at least 200 nucleotides that do not code for proteins. Nowadays, lncRNAs and circRNAs are known to play crucial roles in many biological processes, including a plethora of diseases including cancer. Growing evidence shows an active presence of lnc- and circRNAs in EVs, generating downstream responses that ultimately affect cancer progression by many mechanisms, including angiogenesis. Moreover, many studies have revealed that some tumor cells promote angiogenesis by secreting EVs, which endothelial cells can take up to induce new vessel formation. In this review, we aim to summarize the bioactive roles of EVs with lnc- and circRNAs as cargo and their effect on cancer angiogenesis. Also, we discuss future clinical strategies for cancer treatment based on current knowledge of circ- and lncRNA-EVs.

## 1. Introduction

### 1.1. Cancer Generalities

Cancer is defined as a group of diseases that are multifactorial in nature and represent a challenge in their diagnosis and treatment due to their etiological diversity [1]. More than 200 types of human cancer have been identified based on the cell or tissue from where they originate, the somatic mutations acquired at any time of the progression of the disease, and the microenvironment influences in which they develop [2]. One of the hallmark features of cancer is its rapid and uncontrolled progression due to mutations that alter the cell cycle and overpass checkpoint regulation between the cell cycle phases, promoting the accumulation of mutations passed down to the progeny [3]. For this rapid progression to occur, the growing tumor has a high demand for nutrients and other components; thus, these cells generate molecular signaling to promote the formation of new blood vessels from preexisting ones, a process denominated angiogenesis [4]. The angiogenesis process is fundamental for cancer to advance locally and facilitate metastasis, and therefore, it has been extensively studied in most cancer types [5,6,7]. Multiple efforts have been made to develop antiangiogenic therapies to halt tumor growth and prevent metastasis [8,9]. Recently, the role of extracellular vesicles (EVs) between vessels in tumor communication has triggered the interest of many researchers.

### 1.2. Extracellular Vesicles

Extracellular vesicles are subcellular structures that are heterogeneous in nature and surrounded by a lipid bilayer membrane that exerts multiple functions in intercellular communication [10]. Based on how they are delivered from the original cell to the extracellular medium, EVs can be released by inward budding of the endosomal membrane or outward budding of the cellular membrane [11]. The recipient cell can then internalize EVs through endocytosis or membrane fusion to unload their contents into the cell cytoplasm (Figure 1) [12]. Since their discovery in the early 1980s, many biomolecules have been identified as cargo in EVs, including proteins, amino acids, signaling lipids, and different genetic molecules like DNA, RNA, and non-coding RNAs, promoting both physiological and pathological processes [13]. Based on their biogenesis, EVs are generally classified into microvesicles and exosomes [10], although some authors suggest a further division into apoptotic bodies and proteasomes [14]. Microvesicles are generated by outward budding of the plasma membrane and range from 50 nm to 1000 nm. In contrast, exosomes are membrane vesicles smaller in size (30–100 nm) and are formed by inward budding of the endosomal membrane to be later secreted by fusion with the cell membrane [14,15]. The role of EVs as cell-to-cell mediators in respiratory disease [16], neurodegenerative disease [17], kidney disease [18], cardiovascular disease [19], and cancer progression and metastasis [20,21,22] has been documented to ameliorate the understanding of the behavior or these subcellular structures.

### 1.3. Long Non-Coding and circRNAs

Long non-coding RNAs (lncRNAs) are a diverse group of RNAs that are not translated into proteins, and they are at least 200 nucleotides in length [24]. Recent advances in genomic sciences through RNA sequencing have offered the identification of lncRNAs performing functions to control chromatin complexes, recruit transcription factors, regulate alternative splicing, affect mRNA translation, sponge micro-RNAs by binding, degrade other RNAs, and serve as scaffolds for protein interactions (Figure 2) [25,26,27]. Evidence suggests an active role of lncRNAs in most physiological processes, and their involvement in disease has been the focus of active research in recent years [28,29]. The involvement of lncRNAs as oncogenes or tumor suppressors in many cancer types has also been documented. However, as new lncRNAs are discovered, the general landscape becomes complicated as the roles they can perform become more complex [30]. For instance, lncRNAs can influence the progression of cancer by promoting metastasis [31], drug resistance [32], epithelial-to-mesenchymal transition (EMT) [33], and angiogenesis [34].

It has recently been shown that some lncRNAs can take a circular shape and join covalently at the ends, and these are called circRNAs [35]. This type of lncRNA can perform similar functions to linear lncRNAs, as sponges to recruit specific miRNAs or as effectors to regulate the expression of certain genes [36]. circRNAs have recently been widely studied, arousing interest due to their stability, since, unlike linear non-coding RNAs, they are difficult to degrade [37].

**Figure 2 ncrna-10-00012-f002:**
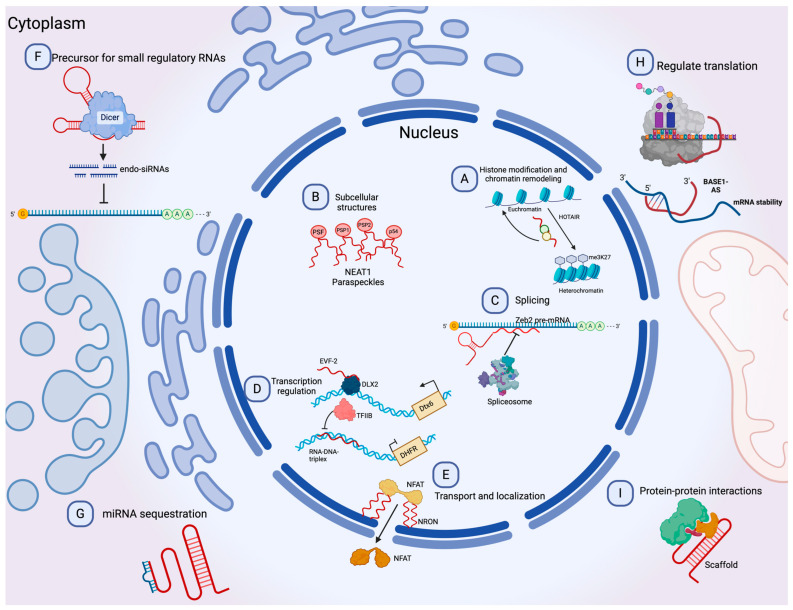
Molecular functions of lncRNAs. (**A**) lncRNAs can guide chromatin complexes, controlling between transcriptionally active euchromatin and silent heterochromatin. (**B**) The recruitment of polymerase II and transcription factors can be inhibited or facilitated by lncRNAs. (**C**) lncRNAs contribute to transcriptome complexity by regulating alternative splicing of pre-mRNAs. (**D**) lncRNAs affect the stability and translation of mRNA by base-pairing with mRNA molecules. (**E**) They influence the expression of miRNAs by binding to them and preventing their function. (**F**) lncRNAs can act as siRNAs and target other RNAs, which subsequently could result in target degradation. (**G**) lncRNAs can join multiple protein factors as flexible scaffolds to interact with or cooperate in protein–protein interactions. (**H**,**I**) The scaffold function is also important for protein activity and localization as well as subcellular structures. Modified from Peña-Flores et al. [38].

Recently, the presence of both coding and non-coding RNA in EVs has motivated research to elucidate RNA’s role in various biological mechanisms in cancer and other diseases [39,40]. This review aims to provide recent evidence on the influence of exosomal lncRNAs on cancer angiogenesis. A systematic screening of papers was performed on PubMed, Google Scholar, Cochrane Library, Web of Science, and EMBASE up to July 2023 for articles matching the following criteria: (long non-coding RNA or lncRNA or ceRNA or circRNA or circular RNA) and (angiogenesis or lymphangiogenesis or vasculogenic mimicry) and (extracellular vesicles or EV or exosome or exosomal). The titles and abstracts were carefully read, and full-text manuscripts relevant to the subject of study were acquired for further analysis.

## 2. Mechanism of Angiogenesis

The angiogenesis process embodies forming new blood vessels from existing vessels in response to physiological and pathological mechanisms [41]. During embryogenesis, the vascular network develops through a combination of vasculogenesis, referred to as the de novo formation of the heart and new blood vessels from stem endothelial cells, namely, angioblasts, and angiogenesis, which expands the initial primitive vascular plexus [42]. Although most blood vessels remain quiescent under physiological conditions, tissue repair and regeneration through wound healing, ovulation, and endometrial thickening throughout the menstrual cycle are based on angiogenesis for proper functioning [43,44,45]. While vascular growth varies depending on where angiogenesis is initiated and the tissue to which they will provide a new blood supply, several mechanisms are common in forming these vessels [46]. In a hypoxic state, the recruitment of cells that promote inflammation; angiogenic growth factor production; degradation of the basement membrane; and endothelial cells (ECs) sprouting, migrating, proliferating, differentiating, and modulating vascular support cells are some of the shared characteristics in angiogenesis [47].

The angiogenic process (Figure 3) comprises several stages involving the sprouting, migration, and proliferation of ECs guided by the vascular endothelial growth factor (VEGF) [48]. Following VEGF stimulation, pericytes from the vessel wall detach, and the basal membrane is weakened by proteolytic degradation. At the same time, ECs adopt an invasive and motile phenotype called tip cells that send out filamentous pseudopodia to guide vascular budding [49]. The cells behind the tip cells are denominated stalk cells, which proliferate to maintain the integrity of the structure and function of the nascent vessels, mainly expanding the vascular lumen [50]. ECs modify their shape by negatively charging glycoproteins on the apical surface to repel each other and open the lumen while redistributing cell-to-cell adhesion to the periphery [51]. For maturation to occur, pericytes must be recruited by the platelet-derived growth factor subunit B (PDGF-B) and angiopoietin 1 (Ang1) signaling along with the strengthening and consolidation of the adhesion between ECs with junctional molecules such as VE-cadherin, while a basement membrane is deposited by tissue inhibitors of metalloproteinases (TIMPs) [52,53].

Under physiological conditions, angiogenesis is strongly regulated by factors ranging from metabolites to hormones [54]. Various molecular pathways have been extensively studied that comprise the angiogenesis process, including the VEGF-VEGFR, Angiopoietin-Tie, Delta-Notch, and Ephrin-Eph [55]. Angiogenesis, vasculogenesis, and lymphangiogenesis are mostly regulated by six VEGF members encoded in the human genome, namely, VEGF-A, VEGF-B, VEGF-C, VEGF-D, VEGF-E, and the placenta growth factor (P1GF) [48,56]. According to their molecular configuration and affinity, the aforementioned factors may bind to different tyrosine kinase receptors VEGFR-1, VEGFR-2, and/or VEGFR-3; VEGF-A binds to VEGFR-2 to contribute to angiogenesis, whereas VEGF-C and VEGF-D enhance lymphatic vessel sprouting by binding to VEGFR-3 [57]. Another group of angiogenesis inducers are the platelet-derived growth factors (PDGF), which induce ECs proliferation and migration by binding to two tyrosine kinase receptors, PDGFR-α and PDGFR-β [58]. Another angiogenesis factor is the fibroblast growth factor (FGF) 2, responsible for inducing metalloproteinase (MMP) secretion to degrade the basement membrane and promote vessel sprouting along with VEGF [59]. Angiopoietin 1, interleukin 8 (IL-8), epidermal growth factor (EGF), and tumor necrosis factor α (TNF-α) also exert a pro-angiogenic effect in ECs through several signaling pathways [60,61,62,63]. Conversely, angiopoietin 2, angiostatin, endostatin, vasostatin, and TIMPs inhibit angiogenesis and play an important role in achieving vascular homeostasis [64,65].

Extracellular vesicles produced by many cellular lineages under specific circumstances can be taken up by ECs to promote and regulate angiogenesis [66]. For instance, a study in endometrial stromal cells (HESCs) found active secretion of EVs during decidualization in a controlled manner by the hypoxia-inducible factor 2 alpha (HIF2α)–Ras-related protein Rab-27B (RAB27B) cascade, revealing a cargo with a variety of growth regulators, signaling molecules, metabolic modulators, and factors that control the expansion and remodeling of ECs [67]. In a myocardial infarction animal model, stem cell-derived small extracellular vesicles (sEVs) loaded with miR-486-5p promoted cardiac angiogenesis via fibroblastic MMP19-VEGFA cleavage signaling [68]. Moreover, Gregorius et al. [69] evaluated the effects of mesenchymal stromal cell (MSC)-derived sEVs on the proliferation, migration, and tube formation of cerebral microvascular ECs. Interestingly, hypoxic preconditioning enhanced angiogenesis and increased post-ischemic endothelial survival by regulating several miRNAs through the uptake of protein-enriched sEVs cargo. Another study demonstrated that HS-1 protein X-1 (HAX1), a major regulator of myeloid homeostasis, was present in EVs secreted by nasopharyngeal carcinoma (NPC) tumors, promoting an angiogenesis phenotype by activating the focal adhesion kinase (FAK) pathway in ECs by increasing the expression level of integrin subunit beta 6 (ITGB6) [70]. Conversely, EVs produced by bone marrow MSCs were found to be enriched in the cluster of differentiation 39 (CD39), TIMP-1, and CD73, inhibiting tumor angiogenesis by targeting the extracellular matrix remodeling and the endothelial cell migration [71].

## 3. Exosomal Long Non-Coding and Circular RNAs in Cancer Angiogenesis

LncRNAs in EV cargo have been demonstrated lately, mainly in cancer [72,73,74]. A recent study launched an online repository of EV long RNAs (exLRs) in diverse human body fluids, comprising 19,643 mRNAs, 15,645 lncRNAs, and 79,084 circRNAs obtained from human blood, cerebrospinal fluid, bile, and urine samples. The database provides novel exLR signatures to help discover new biomarkers that could aid in diagnosing and treating many diseases [75]. Based on available recent research, Casado-Díaz et al. [76] concluded that lncRNAs and other RNAs included in MSC-derived EVs can be applied in chronic skin ulcers to improve accelerated healing and decrease scar formation due to immunosuppressive and immunomodulatory properties. Conversely, in a diabetic wound-healing animal model, upregulated lncRNAs packed in EVs from fibroblasts enhanced keratinocyte MMP-9 expression to induce collagen degradation, delaying wound healing [77]. Recently, the long non-coding repressor of NFAT (NRON) was detected in BMSC-derived EVs, inhibiting osteoclast differentiation and osteoporotic bone loss in vitro and in vivo [78].

In tumors, the high rate of cell proliferation forces the formation of new blood vessels [79]. However, in most cases, these blood vessels are dilated, tortuous, and immature, leading to excessive permeability and increased hypoxia [80]. In addition, vascular disorganization causes heterogeneity in the tumor blood vessel network, creating highly vascularized tumor areas and other hypoxic areas with low vascular density [47]. Thus, hypoxia becomes a major driver of tumor angiogenesis, along with other mechanisms promoted by activated oncogenes or loss of tumor suppressor genes, in which lncRNAs play an important role, mainly through acting as competing endogenous RNAs for miRNAs [81]. Similarly, circRNAs have been extensively studied in cancer, elucidating important roles in tumor development, growth, and angiogenesis [82]. For instance, VEGFR-related pathways have been linked to circRNAs by affecting tumor angiogenesis by sponging miRNAs [83,84]. The landscape of exosomal lnc- and circRNAs in angiogenesis in cancer is summarized in Table 1 and Figure 4.

### 3.1. Bone Malignancies

Some studies have been dedicated to studying exosomal lncRNAs in bone malignancies. LncRNA Opa-interacting protein 5-antisense 1 (OIP5-AS1) was found to be overexpressed in exosomes secreted by osteosarcoma cells, increasing angiogenesis in tubule formation assays by mechanistically sponging miR-153 and increasing the autophagy-related 5 protein (ATG5) [85]. Interestingly, serum samples from osteosarcoma patients could transfer via EVs the myocardial infarction-associated transcript (MIAT), promoting the proliferation of osteosarcoma cell lines and angiogenesis in HUVECs by sponging miR-613 and upregulating G protein-coupled receptor 158 (GPR158) [86]. In an in vitro and animal model, BMSC-EVs carried the non-coding RNA activated by DNA damage (NORAD) into osteosarcoma cells and upregulated CREB-binding protein (CREBBP) by sponging miR-877-3p to promote proliferation, invasion, migration, and angiogenesis [87]. Another lncRNA called Ewing sarcoma-associated transcript 1 (EWSAT1) was found to regulate osteosarcoma-induced angiogenesis via two mechanisms: (1) by increasing in sensitivity/reactivity of vascular endothelial cells triggered by exosomes carrying EWSAT1, and (2) by increasing angiogenic factors secretion [88]. Moreover, exosomes secreted by chondrosarcoma cells were loaded with the receptor activity-modifying protein 2 antisense 1 (RAMP2-AS1). They could enhance HUVECs proliferation, migration, and tube formation by acting as a ceRNA for miR2355-5p to regulate VEGFR2 expression. In addition, the overexpression of RAMP2-AS1 in the serum of chondrosarcoma patients was demonstrated to be closely related to local invasiveness, distant metastasis, and poor prognosis [89].

### 3.2. Esophageal, Gastric, and Colorectal Cancers

Some of the most prevalent tumors of the gastrointestinal (GI) tract have been explored regarding the role played in angiogenesis by exosomes loaded with different lncRNAs and circRNAs. For instance, the exosomal lncRNA family with sequence similarity 225 member A (FAM225A) was highly expressed in esophageal squamous cell carcinoma (ESCC), upregulating neuropilin and tolloid-like 2 (NETO2) and forkhead box P1 (FOXP1) expression by sponging miR-206 to accelerate tumor progression and angiogenesis [93].

In gastric cancer patients, exosomal circ-SHKBP1 was overexpressed in tumor and blood samples. When the exosomes were isolated and exposed to different cell lines, cells showed a promoted proliferation, invasion, migration, and angiogenesis rate by mechanistically regulating the miR-582-3p/HUR/VEGF axis and suppressing heat shock protein 90 (HSP90) degradation [116]. Similarly, 30 blood samples and tissues from gastric cancer patients were taken to analyze circ-FCH and mu domain-containing endocytic adaptor 2 (FCHO2). It was found that circ-FCHO2 up-modulation led to a poor outcome, while circ-FCHO2 silencing weakened the proliferation, invasion, angiogenesis, and stem cell characteristics, presumably by activating the Janus kinase 1 (JAK1)/signal transducer and activator of transcription 2 (STAT2) pathway via sponging miR-194-5p [117]. Conversely, by acting as a miR-587 sponge to adjust the expression of the sclerostin domain-containing 1 (SOSTDC1), circ-0001190 overexpression inhibited cell viability, proliferation, angiogenesis, migration, and invasion of gastric cancer cell lines [114]. Moreover, circ-0044366 was highly expressed in gastric cancer and impaired the proliferation, migration, and tube formation of HUVECs by exosomal communication by acting as miR-29a ceRNA and regulating the VEGF pathway [115].

In colorectal cancer (CRC), tumor growth, angiogenesis, and liver metastasis were suppressed by exosomal circ-fibronectin type III domain-containing 3B (FNDC3B) overexpression by acting via the miR-97-5p/TIMP3 pathway [98]. Similarly, exosomes derived from lncRNA adenomatous polyposis coli (APC1)-silenced CRC cells promoted angiogenesis by activating the mitogen-activated protein kinase 1 (MAPK) pathway in endothelial cells, while enforced APC1 was sufficient to inhibit CRC growth, metastasis, and tumor angiogenesis by suppressing exosome production [99]. Interestingly, exosomes loaded with circ-tubulin gamma complex component 4 (TUBGCP4) derived from CRC cells enhanced vascular endothelial cell migration and tube formation via inducing filopodia formation and endothelial cell tipping by upregulating the pyruvate dehydrogenase kinase 2 (PDK2) to activate the AKT serine/threonine kinase 1 (AKT) signaling pathway and by sponging miR-146b-3p [100]. A very interesting study by Zhi et al. [125] compared EVs derived from the b-Raf proto-oncogene (BRAF) wild-type CRC and the BRAF^V600E^ mutant patients to find the overexpression of 13 lncRNAs and downregulation of 22 lncRNAs in exosomes from the BRAF^V600E^ mutation type. This difference showed a higher microvascular and micro-lymphatic vessel density of the BRAF^V600E^ mutant CRC tissues.

### 3.3. Liver and Pancreatic Cancers

LncRNA-loaded exosomes from tumors from other organs related to the GI tract have also shown some relationship with tumor angiogenesis. You et al. [101] reported high levels of Linc-00161 in serum-derived exosomes from hepatocellular cancer (HCC) patients and the supernatants of HCC cell lines, which are associated with poor survival. Mechanistically, Linc-00161 promoted angiogenesis in HUVECs by inhibiting miR-590-3p and activating the Rho-associated coiled-coil-containing protein kinase 2 (ROCK2) axis. In an in vitro study, exosomes with lncRNA H19 were released by CD90+ HCC cells and modulated endothelial cells, promoting an angiogenic phenotype and cell-to-cell adhesion [103]. Similarly, lncRNA ubiquitin-conjugating enzyme E2 C pseudogene 3 (UBE2CP3) was overexpressed in HCC EVs. It promoted HUVEC proliferation, migration, and tube formation via the activation of the ERK/HIF-1α/p70S6K/VEGFA signaling cascade, promoting HCC tumorigenicity [102]. In another study, exosomal circ-100388 affected the cell proliferation, angiogenesis, permeability, and vasculogenic mimicry formation ability of HUVECs and HCC tumor metastasis [104].

In cholangiocarcinoma (CCA), the cholangiocarcinoma-associated circular RNA 1 (circ-CCAC1) from CCA-derived EVs was transferred to endothelial monolayer cells, disrupting endothelial barrier integrity and inducing angiogenesis. Interestingly, circ-CCAC1 increased cell leakiness by sequestering the enhancer of zeste 2 polycomb repressive complex 2 subunit (EZH2) gene, thus elevating the SH3 domain-containing GRB2 like 2, endophilin A1 (SH3GL2) expression to reduce levels of intercellular junction proteins [124].

In pancreatic cancer, the expression levels of the lncRNA urothelial cancer-associated 1 (UCA1) in exosomes derived from the serum of patients were associated with poor survival, promoting angiogenesis and tumor growth by acting as a ceRNA of miR-96-5p, relieving the repressive effects on the expression of its target gene angiomotin like 2 (AMOTL2) [90]. Moreover, the exosomal small nucleolar RNA host gene 11 (SNHG11) promoted cell proliferation, migration, and angiogenesis in pancreatic cancer cell lines but impeded cell apoptosis via sponging miR-324-3p to upregulate VEGFA expression [126].

### 3.4. Renal and Bladder Cancers

Some urinary system tumors have observed a relationship between lncRNA-loaded EVs and angiogenesis. In renal cell carcinoma (RCC), RCC-derived exosomes had an lncRNA Ars operon (ARSR) that promoted macrophage polarization, cytokine release, phagocytosis, angiogenesis, and tumor development by sponging miR34/miR-449 and upregulating the signal transducer and activator of transcription 3 (STAT3) pathway [108]. Similarly, RCC-derived exosomal circular scaffold attachment factor B2 (circ-SAFB2) facilitated the progression, invasion, angiogenesis, and metastasis of RCC by inducing the polarization of M2 macrophages through the miR-620/JAK1/STAT3 axis [109]. Conversely, exosomal circular spire type actin nucleation factor 1 (circ-SPIRE1) suppressed angiogenesis and vessel permeability through regulating ELAV-like RNA-binding protein 1-mRNA, binding and upregulating polypeptide N-Acetylgalactosaminyltransferase 3 (GALNT3) and KH domain RNA-binding protein (QKI) expression [110].

In bladder carcinoma (BCa), exosomal brain cytoplasmic RNA 1 (BCYRN1) promoted the tube formation and migration of human lymphatic endothelial cells (HLECs), upregulating the Wnt family member 5A (WNT5A) gene expression by inducing hnRNPA1-associated H2K4 trimethylation in WNAT5a promoter, which activated Wnt/β-catenin signaling to facilitate the secretion of VEGF-C in BCa [111]. Moreover, lymph node metastasis-associated transcript 2 (LNMAT2)-loaded exosomes from BCa tissues and blood samples stimulated the tube formation and migration of HLECs and enhanced tumor lymphangiogenesis and lymph node metastasis by upregulation of prospero homeobox 1 (PROX1) gene expression by recruitment of hnRNPA2B1 and increasing H3K4 trimethylation [112]. Comparably, BCa cell-secreted EVs mediated intercellular communication with HLECs through the transmission of the small nucleolar RNA host gene 16 (ELNAT1) and promoted lymphangiogenesis by inducing the ubiquitin-conjugating enzyme E2 (UBC9) gene overexpression to catalyze the small ubiquitin-like modifier (SUMO) binding of hnRNPA1 at the lysine 113 residue [113].

### 3.5. Nasopharyngeal and Lung Cancers

LncRNAs in exosomes derived from nasopharyngeal squamous cell carcinoma (NPSCC) and their relationship with tumor angiogenesis have been mildly explored. In serum samples from newly diagnosed NPSCC patients, the long intergenic non-protein-coding RNA, regulator of reprogramming (linc-ROR), was substantially expressed in exosomes that could be taken up by HUVECs, increasing proliferation, migration, and angiogenesis in vitro by mechanistically upregulating the p-AKT/p-VEGFR2 pathway [91]. Similarly, lncRNA colon cancer-associated transcript 2 (CCAT2) was found in EVs derived from NPSCC patients, promoting HUVEC proliferation and angiogenesis promotion [92].

In non-small cell lung carcinoma (NSCLC), NSCLC cells secreted exosomes with melanotransferrin antisense 1 (MFI2-AS1) to induce tube formation by HUVECs, promoting angiogenesis and metastasis by sponging miR-107, which in turn activated the PI3K/AKT pathway [105]. Similarly, high EV Linc-p21 was found in NSCLC blood samples from tumor-draining pulmonary veins before tumor surgical resection. EVs with Linc-p21 were taken up by HUVECs and promoted tube formation and enhanced tumor cell adhesion to endothelial cells by sponging miR-23a, miR-146bv, miR-330, and miR-494 [106]. In contrast, GAS5 was lowly expressed in human lung cancer tissues, lung cancer cells, and cell culture supernatant exosomes. The exosomes of lung cancer cells containing high GAS5 levels inhibited HUVECs proliferation and tube formation, increasing their apoptosis by sponging miR-29-3p and upregulating phosphatase and tensin homolog (PTEN) and inhibiting PI3K/AKT phosphorylation [107].

### 3.6. Glioma and Gliobastoma

A few studies have reported evidence of the role of exosomal lncRNAs in glioma and glioblastoma angiogenesis. An in vitro study with glioma cell lines demonstrated that HUVECs can take up exosomal CCAT2 to promote migration, proliferation, tubular-like structure formation, and arteriole formation [94]. Similarly, the POU class 3 homeobox 3 (POU3F3) was upregulated in glioma tissue. When human brain microvascular endothelial cells (HBMVECs) were treated with exosomes loaded with POU3F3, they exhibited better migration, proliferation, tubular-like structure formation, and arteriole formation. Mechanistically, POU3F3 was shown to upregulate bFGF, bFGFR, VEGFA, and Angio [95]. Moreover, cell line A172 was cultured to demonstrate that EVs loaded with the HOX transcript antisense RNA (HOTAIR) had a pro-angiogenic activity in HBMVECS via VEGFA [96]. In glioblastoma, lncRNA HIF1A antisense RNA 2 (AHIF) was found upregulated in tissue samples, and when cultured with glioblastoma cell lines, exosomal AHIF regulated factors associated with migration and angiogenesis [97].

### 3.7. Other Cancer Types

In ovarian cancer, an in vitro study revealed that lncRNA activated by TGF-β (ATB) promoted viability and angiogenesis of HUVECs by sponging miR-204-3p and thus upregulating TGFβ-R2 [127]. Similarly, elevated serum exosomal metastasis-associated lung adenocarcinoma transcript 1 (MALAT1) promoted angiogenesis and was highly correlated with an advanced and metastatic phenotype of epithelial ovarian cancer [118]. Another study demonstrated taurine-upregulated 1 (TUG1) overexpression in human cervical cancer cell lines. When TUG1 was depleted, the exosome-mediated pro-angiogenic potential of HUVECs was impaired by modulating angiogenesis-related genes like VEGFA, MMP9, TGFβ, IL-8, and bFGF [119]. In breast cancer cell lines, the metadherin (MTDH) gene improved cell viability and angiogenesis in endothelial cells. The molecular cascade was promoted by exosomal circular homeodomain-interacting protein kinase 3 (circ-HIPK3), which sponged miR-124-3p and in turn upregulated MTDH [120]. Liu et al. Campo [121] demonstrated that exosomal overexpression of the FYVE, RhoGF, and PH domain-containing 5 antisense 1 (FGD5-AS1) enhanced the proliferation, migration, angiogenesis, and permeability of HUVECs by regulating the endothelial miR-6838-5p/Vav guanine nucleotide exchange factor 2 (VAV2) axis. A total of 25 peripheral blood samples from 20 multiple myeloma patients and 5 matched healthy controls showed overexpression of the circular ATPase phospholipid-transporting 10A (circ-ATP10A) in the multiple myeloma samples, mechanistically acting as a sponge of several miRNAs to consequently regulate the expression of downstream VEGFB, HIF1A, PDGFA, and FGF [122].

## 4. Clinical Relevance and Future Perspectives

In recent years, a plethora of studies have demonstrated the role played by lncRNAs and circRNAs in many molecular and cellular processes, ranging from early development to complex diseases such as cancer [128,129,130]. Although the intracellular expression of these non-coding RNAs has been documented, the presence of DNA and RNA fragments in extracellular vesicles and exosomes has also been discovered, demonstrating the ability of cells to transmit information not only to their environment and surrounding cells but also to distant areas through fluids like plasma, bile, and urine [131,132,133]. For instance, exosomes derived from serum from pancreatic cancer patients were associated with poor survival when loaded with UCA1 [90]. Similarly, elevated serum exosomal MALAT1 was an independent predictive factor for ovarian cancer overall survival [118]. In the sera of chondrosarcoma patients, exosomal RAMP2-AS1 was closely related to local invasiveness, distant metastasis, and poor prognosis [89]. Serum-derived exosomes loaded with linc-00161 from hepatocellular cancer patients were significantly associated with poor survival [101], and high EV linc-p21 levels in blood were associated with shorter time to relapse and shorter overall survival in lung cancer [106]. The presence of cholangiocarcinoma-derived EVs loaded with circ-CCAT1 was detected in bile samples from perihilar and distal CCA patients, demonstrating a worse overall prognosis [124]. Interestingly, 63% of patients with bladder cancer evaluated as lymph node metastasis (LN)-negative by CT were correctly predicted as being LN-positive by the detection of urinary EV-mediated ELNAT1 [113].

EV-mediated lncRNAs are promising early diagnostic biomarkers and potential therapeutic targets in many diseases. As the relationship between exosomal lnc- and circRNAs and their role in angiogenesis is further studied, tools may be developed for early and accurate diagnosis of diseases such as diabetes and cancer, establishing therapeutic pathways that promote a better prognosis for patients with these and other diseases. The sensitivity and specificity for the detection of circulating RNAs must be improved to apply these techniques in regular clinical practice.

## 5. Conclusions

Exosomes have recently been the subject of study due to the molecular cargo they possess since they apparently function as intercellular communication mechanisms. Among the components that can generate changes in other cells are lnc- and circRNAs, which can influence fundamental processes for functional and pathological development, such as angiogenesis. As the roles of exosomal lnc- and circRNAs in angiogenesis are elucidated, diagnostic and prognostic tools can be developed to improve advanced systemic disease therapies.

## Figures and Tables

**Figure 1 ncrna-10-00012-f001:**
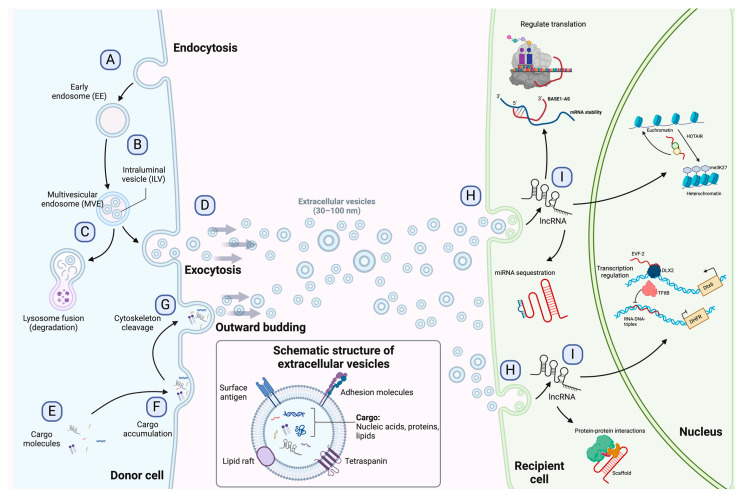
Exosome biogenesis, release to the extracellular environment, and uptake by the recipient cell. (**A**) Exosome biogenesis begins with early endosome formation during endocytosis. (**B**) Early endosomes are then matured into late endosomes, generating multiple intraluminal vesicles (ILVs) by the inward budding of endosomal membranes. (**C**) The accumulation of ILVs leads to the formation of multivesicular endosomes (MVEs), and proteins and nucleic acids produced by the donor cell can be sorted into exosomes during MVE formation. (**D**) Exosomes are released into the extracellular environment by fusing MVEs with the cellular membrane. (**E**–**G**) Microvesicles arise from the outward budding and shedding of the plasma membrane. (**H**) Extracellular vesicles are taken up by the recipient cell by direct fusion, receptor-mediated fusion, or endocytosis. (**I**) Exosomal lncRNAs can be subsequently delivered to the recipient cell to exert regulatory effects as sponges for miRNAs, protein scaffolds, transcription and translation regulators, and chromatin activators. The detailed functions of lncRNAs are depicted in Figure 2. Figure 1 is modified from Wang et al. [23].

**Figure 3 ncrna-10-00012-f003:**
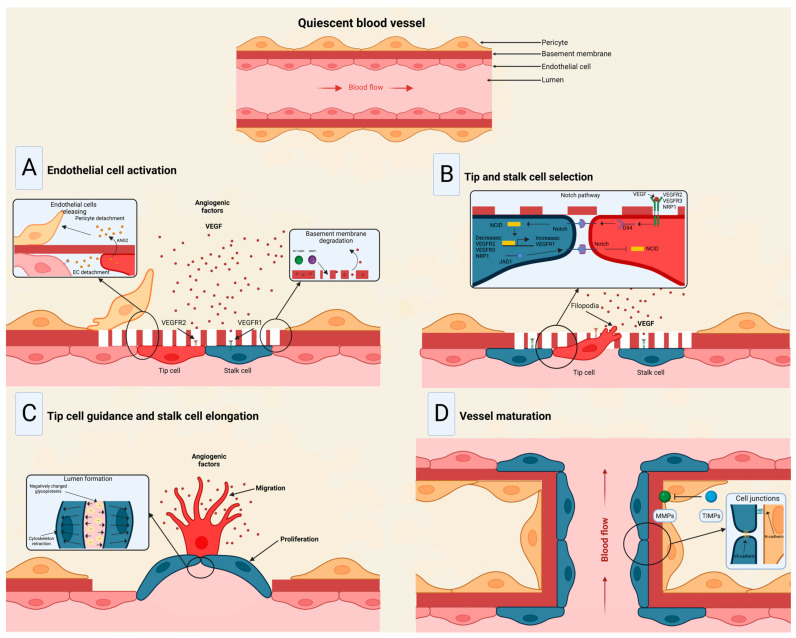
Stages of the angiogenic process. (**A**) Angiogenic signals, such as VEGF, promote pericyte detachment from the basement membrane and weaken the extracellular matrix. (**B**) Endothelial cells display characteristic phenotypes after VEGF stimulation: migratory tip cells or proliferating stalk cells. (**C**) Attractive and repulsive forces control endothelial cells, forming a vessel lumen to initiate blood flow. (**D**) PDGF-B and Ang1 signaling lead to pericyte recruitment, while junctional molecules consolidate EC–EC adhesion. Modified from Viallard et al. [47].

**Figure 4 ncrna-10-00012-f004:**
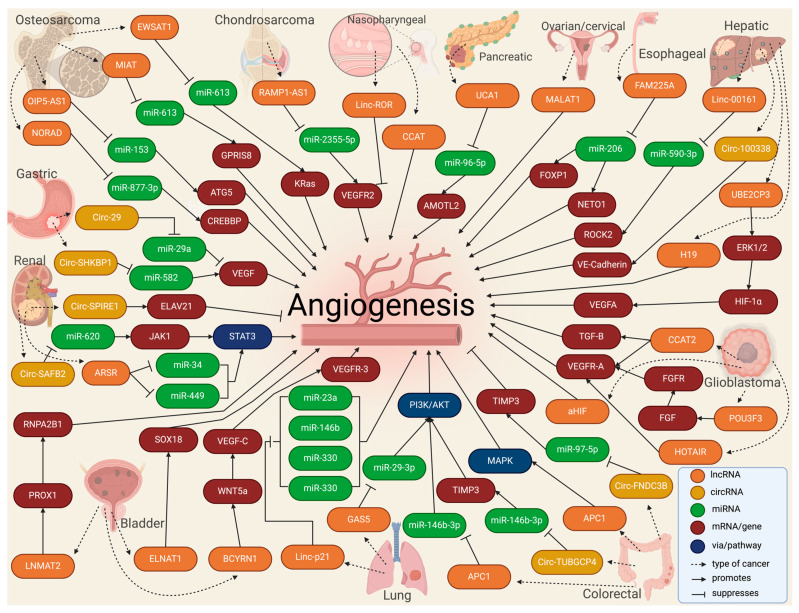
Molecular landscape of exosomal lnc- and circRNAs in angiogenesis in cancer.

**Table 1 ncrna-10-00012-t001:** The landscape of exosomal lnc- and circRNAs in angiogenesis in cancer.

Lnc/circRNA	Molecular Target	Donor Cells	Recipient Cells	Effect	Reference
**OSTEOSARCOMA**					
OIP5-AS1	miR-153/ATG5	HOS	HUVECs	Promotes	Li 2021 [85]
MIAT	miR-613/GPR158	U2OS, MG63, and 293T	HUVECs	Promotes	Wang 2022 [86]
NORAD	miR-877-3p/CREBBP	143B, MG-63, Saos2, HOS, and U20S	Osteosarcoma cells	Promotes	Feng 2022 [87]
EWSAT1	miR-326/KRas	143B, MNNG/HOS, MG63, U20S	BMSCs, HMEC-1	Promotes	Tao 2020 [88]
**CHONDROSARCOMA**					
RAMP2-AS1	miR-2355-5p/VEGFR2	SW1353	HUVECs	Promotes	Cheng 2020 [89]
**PANCREATIC**					
UCA1	miR-96-5p/AMOTL2	PANC-1, MIA PaCa-2, BxPC-3, Aspc-1, Sw1990	HUVECs, HEK293T	Promotes	Guo 2020 [90]
**NASOPHARYNGEAL**					
Linc-ROR	p-AKT/p-VEGFR2 pathway	CNE2	HUVECs	Promotes	Zhang 2022 [91]
CCAT2	NR	CNE2, NP69	HUVECs	Promotes	Zhou 2020 [92]
**ESOPHAGEAL**					
FAM225A	miR-206/NETO2 and FOXP1	ECA109, TE-1, KYSE150, KYSE140	HET-1A, HUVECs	Promotes	Zhang 2020 [93]
**GLIOMA/GLIOBLASTOMA**					
CCAT2	VEGF-A and TGF-B	A172, U87-MG, U251, T98G	HUVECs	Promotes	Lang 2017 [94]
POU3F3	bFGF/bFGFR/VEGFA	A172, U87-MG, U251, T98G	HBMECs	Promotes	Lang 2017 [95]
HOTAIR	VEGFA	A172	HBMVECs	Promotes	Ma 2017 [96]
aHIF	NR *	U87-MG, U251-MG, A172, T98G	HUVECs	Promotes	Dai 2019 [97]
**COLORECTAL**					
CircFNDC3B	miR-97-5p/TIMP3	LoVo, SW480, SW602, HCT116	HUVECs	Suppresses	Zeng 2020 [98]
APC1	MAPK pathway	HTC116, DLD-1, SW480, LoVo, SW116	HEK293T, HUVECs	Promotes	Wang 2019 [99]
CircTUBGCP4	miR-146b-3p/PDK2/Akt	SW480	HEK297T	Promotes	Chen 2023 [100]
**LIVER/HEPATOCELLULAR**					
LINC00161	miR-590-3p/ROCK2 axis	Huh-7, HCCLM3, MHCC-97L, MHCC-97H	WRL-68, HUVECs	Promotes	You 2021 [101]
UBE2CP3	ERK1/2/HIF-1α/VEGFA	HepG2, SMMC-7721	HUVECs	Promotes	Lin 2018 [102]
H19	NR *	Huh-7, Sk-Hep	HUVECs	Promotes	Conigliaro 2015 [103]
Circ100338	VE-Cadherin	Hep3B, HLE, Huh-7, BEL7402, SMCC7721, MHCC97L, HCCLM3, MHCC97H, HCCLM6	HUVECs	Promotes	Huang 2020 [104]
**LUNG**					
MFI2-AS1	miR-107/PI3K/AKT pathway	PC9, A549, H1299	HUVECs	Promotes	Xu 2023 [105]
LincRNA-p21	miR-23a, miR-146b, miR-330, and miR-494	H23, HCC44	HUVECs	Promotes	Castellano 2020 [106]
GAS5	miR-29-3p/PI3K/Akt	16HBE, A549, H1299, 95D	HUVECs	Promotes	Cheng 2019 [107]
**RENAL CELL**					
ARSR	miR-34 and miR-449 to upregulate STAT3 pathway	Caki-1, ACHN, 786-O	NR *	Promotes	Zheng 2022 [108]
CircSAFB2	miR-620/JAK1/STAT3 axis	A498, 786-O, Caki-1, Caki-2, 769-P, ACHN	THP-1	Promotes	Huang 2022 [109]
CircSPIRE1	ELAVL1 protein	NR *	NR *	Suppresses	Shu 2023 [110]
**BLADDER**					
BCYRN1	WNT5a/VEGF-C/VEGFR3	T24, 5637, SVHUC-1	HLECs, HDLECs, HUVECs	Promotes	Zheng 2021 [111]
LNMAT2	PROX1/RNPA2B1/H3K4	UM-UC-3, 5637, T24	HLEC, SV-HUC-1	Promotes	Chen 2020 [112]
ELNAT1	SOX18	UM-UC-1, RT112, RT4, UM-UC-3, T24, 5637	HLEC, SV-HUC-1	Promotes	Chen 2021 [113]
**GASTRIC**					
Circ0001190	miR-587/SOSTDC1	NR *	NR *	Suppresses	Liu 2022 [114]
Circ29	miR-29a/VEGF pathway	SGC-7901, MGC-803	HUVECs, HEK297T	Suppresses	Li 2021 [115]
CircSHKBP1	miR-582/HUR/VEGF	AGS, HGC27, BGC823 MGC803, GES1	HUVECs, HEK293T	Promotes	Xie 2020 [116]
CircFCHO2	miR-194-5p/JAK1/STAT3 pathway	NR *	NR *	Promotes	Zhang 2022 [117]
**OVARIAN**					
MALAT1	NR *	SKOV3, HO8910	SKOV3.ip1, HO8910.PM	Promotes	Qiu 2018 [118]
**CERVICAL**					
TUG1	VEGF-A, MMP-9, IL-8	HeLa, CaSki	HUVECs	Suppresses	Lei 2020 [119]
**BREAST**					
CircHIPK3	miR-124-3p/MTDH	NR *	NR *	Promotes	Shi 2022 [120]
**THYROID**					
FGD5-AS1	miR-6838-5p/VAV2 axis	SW1736, KAT18	HUVECs	Promotes	Liu 2022 [121]
**MULTIPLE MIELOMA**					
CircATP10A	miR-66758-3p, miR-3977, miR-6804-3p, miR-1266-3p, miR-3620-3p	NR *	NR *	Promotes	Yu 2022 [122]
**ALCOHOL-INDUCED TUMOR**					
HOTAIR and MALAT1	NR *	NR *	HUVECs, HDMECs	Promotes	Lamichhane 2017 [123]
**CHOLANGIOCARCINOMA**					
CircCCAC1	EZH2/SH3GL2	CCA cells	HUVECs	Promotes	Xu 2021 [124]

* NR: not reported.

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
