# Peer review of "Functional Relevance of Extracellular Vesicle-Derived Long Non-Coding and Circular RNAs in Cancer Angiogenesis"

_ncrna, 2024, doi:10.3390/ncrna10010012_

Round 1

Reviewer 1 Report

Comments and Suggestions for Authors

Overall, the review is well-written and logically presented. However, there are some minor errors that should be clarified or revised. I have also provided suggestions for improving the manuscript before considering it for publication.

- Firstly, on line 74, it is mentioned that Figure 2 was modified from reference 23. However, in the figure legend (line 99), it is claimed to be modified from reference 35. Please ensure consistency in referencing.

- Regarding Table 1, I would recommend placing the heading on the same page as the table to enhance readability. Additionally, providing information about the donor and recipient cells would greatly aid the reader's understanding of the involvement of EVs in angiogenesis.

- In Figure 4, we noticed that the landscapes are categorized by organs. However, the categorization is currently indicated by the same background colour, which can be confusing. Please consider using dashed lines or different background colours to differentiate between organs.

- On line 349, the claim that there is little evidence appears subjective. It would be more appropriate to present the available evidence objectively or rephrase the statement accordingly.

- Are the cancer types mentioned primarily metastasizing via angiogenesis? It would be valuable to discuss the potential involvement of lymphangiogenesis in addition to angiogenesis, considering that lymphangiogenesis was included in the search terms, but the discussion is limited.

Author Response

Thank you very much for taking the time to review this manuscript. We have included the detailed responses below and the revisions/corrections highlighted/in track changes in the re-submitted files.

- Firstly, on line 74, it is mentioned that Figure 2 was modified from reference 23. However, in the figure legend (line 99), it is claimed to be modified from reference 35. Please ensure consistency in referencing.

Thank you for your comment. We have reviewed the references in question, and we realize there is confusion in the wording since the modification of reference 23 refers to Figure 1. To avoid further confusion, we have modified the text to “Figure 1 is modified from Wang et al (23)”. 

- Regarding Table 1, I would recommend placing the heading on the same page as the table to enhance readability. Additionally, providing information about the donor and recipient cells would greatly aid the reader's understanding of the involvement of EVs in angiogenesis.

Thank you for your kind comment and idea. The table heading has been placed on the same page. Additionally, we have added columns “Donor cells” and “Recipient cells” to improve the reader’s understanding.

- In Figure 4, we noticed that the landscapes are categorized by organs. However, the categorization is currently indicated by the same background colour, which can be confusing. Please consider using dashed lines or different background colours to differentiate between organs.

Thank you for your kind observation. Since it is a crowded figure already and background color or dashed lines are difficult to place without further confusion, we have added small arrows to indicate the lnc- or circ-RNA related to the specific type of cancer.

- On line 349, the claim that there is little evidence appears subjective. It would be more appropriate to present the available evidence objectively or rephrase the statement accordingly.

Thank you for your valuable observation. We have rephrased the paragraph, presenting only the evidence that is available.

- Are the cancer types mentioned primarily metastasizing via angiogenesis? It would be valuable to discuss the potential involvement of lymphangiogenesis in addition to angiogenesis, considering that lymphangiogenesis was included in the search terms, but the discussion is limited.

Thank you for your kind comments. Although our search did indeed include terms related to lymphangiogenesis, unfortunately, few studies were found that mentioned it broadly. However, the information found was captured in sections such as those corresponding to lines 105, 150, and 312-317, especially in kidney and bladder cancer studies.

Reviewer 2 Report

Comments and Suggestions for Authors

The whole manuscript has been well prepared, well written. It is a nice review. I have the following suggestions for them to further improve this manuscript:

 1.  The authors combined LncRNAs with circRNAs. It seems that they mainly focus on LncRNAs. This may induce some confusion. They may need point out their differences in cancer angiogenesis and point out why they mixed two different RNA types in this review.

2.   They showed that some LncRNAs or circRNAs may serve as biomarkers or early therapeutic targets in Table 1. It may be a good idea for them to read these papers carefully and clearly point out which of these LncRNAs or circRNAs have been used or can be used as biomarkers or therapeutic targets

3.   In the abstract part, the authors pointed out that tumor cells promote angiogenesis by secreting EVs, which endothelial cells can take up to induce new vessel formation. This may be not completely right because some tumor cells promote angiogenesis by other mechanisms, not just secreting EVs.

Author Response

Thank you very much for taking the time to review this manuscript. Please find the detailed responses below and the revisions/corrections highlighted/in track changes in the re-submitted files.
  1. The authors combined LncRNAs with circRNAs. It seems that they mainly focus on LncRNAs. This may induce some confusion. They may need point out their differences in cancer angiogenesis and point out why they mixed two different RNA types in this review.

Thank you for your kind observation. Although circRNAs are molecularly closed RNAs, they are considered a type of lncRNAs, so that is why we decided to include them in the review as a complement to the information and to cover more on the topic. To better clarify and explain the inclusion of circRNAs in the review topic, we add the following paragraph in the introduction (lines 88-94):

“It has recently been shown that some lncRNAs can take a circular shape and join covalently at the ends, being called circRNA [35]. This type of lncRNA can perform similar functions to linear lncRNAs, as sponges to recruit specific miRNAs or as effectors to regulate the expression of certain genes [36]. circRNAs have recently been widely studied, arousing interest due to their stability, since, unlike linear non-coding RNAs, they are difficult to degrade [37].”

In addition, lines (215-218) were added to mention the role of circRNAs in cancer and angiogenesis briefly.

“Similarly, circRNAs have been extensively studied in cancer, elucidating important roles in tumor development, growth, and angiogenesis [82]. For instance, VEGFR-related pathways have been linked to circRNAs by affecting tumor angiogenesis by sponging miRNAs [83, 84].”

The role of EVs-circRNAs identified in various studies on their function in cancer angiogenesis is mentioned in the different sections, such as lines 253, 258-263, 268, 273, 292, 292-297, 311-313, etc

  1. They showed that some LncRNAs or circRNAs may serve as biomarkers or early therapeutic targets in Table 1. It may be a good idea for them to read these papers carefully and clearly point out which of these LncRNAs or circRNAs have been used or can be used as biomarkers or therapeutic targets

Thank you for your comment and suggestion. Section 4 mentions several non-coding RNAs used in various studies as potential biomarkers in fluids such as blood and bile, in their association with cancer stages, their potential for invasion, metastasis, and poor prognosis. In addition to the evidence presented, the authors did not identify other lncRNAs or circRNAs identified as biomarkers or therapeutic targets in EVs.

  1. In the abstract part, the authors pointed out that tumor cells promote angiogenesis by secreting EVs, which endothelial cells can take up to induce new vessel formation. This may be not completely right because some tumor cells promote angiogenesis by other mechanisms, not just secreting EVs.

Thank you for your kind comment. We have rephrased the part of the abstract where it can be misinterpreted by generalizing that tumor cells promote angiogenesis by secreting EVs. Instead, it has been put that "many studies have revealed that some tumor cells promote angiogenesis by secreting EVs."

Reviewer 3 Report

Comments and Suggestions for Authors

Pena-Flores et al. provides a systematic review of articles published until July 2023 on the topic of “Functional Relevance of Extracellular Vesicle-Derived Long Non-Coding and Circular RNAs in Cancer Angiogenesis”. The study provides a nice account of existing knowledge on how angiogenesis may be modulated by lncRNAs and circRNAs cargoed in exosomes within the concept of cancer biology. I believe that researchers would benefit from this systematic review. I suggest the following points for consideration to improve the quality of the manuscript.

1. Gene names should be written out in full in their first use, followed by abbreviation. Please check the whole manuscript for this point.

2. Please follow the common use of “lncRNAs” and “circRNAs” instead of “lnc-“ or “circ-RNAs” throughout the manuscript.

3. Please ensure that the words like “in vitro”, “in vivo”, “de novo” are italicized.

4. Introduction cover three major concepts, namely cancer, exosomes and ncRNAs. It might be better to use subtitles.

5. Although the title covers the clinical relevance of “lncRNAs” and “circRNAs”, the authors describe only lncRNAs in Introduction (lines 75-99). It would be nice to describe “circRNAs” and perhaps touch on the ceRNA hypothesis.

6. Figure 2: This figure shows the molecular function(s) of lncRNAs not “classification of lncRNAs”. Please either replace the figure or the title.

7. Figure 3: The text on the figure is too small to read. Also, I wonder whether the authors have taken permission from the authors of “Ref. 44”.

8. Please be consistent in the use of numbers (e.g. separation of 5-digit numbers preferably with a comma—please see the line 188).

9. Figure 4: Although this is a highly informative figure, it is challenging to track the tissue source of lncRNA/circRNA. Some of the unnecessary components may be omitted to highlight the essential message. Additionally, the resolution should be improved.

10. Clinical relevance and future perpectives contains only the relevance of lncRNAs without any mention of circRNAs. Please indicate how exosome-cargoed circRNAs can be of relevance.

Comments on the Quality of English Language

Minor editing is sufficient.

Author Response

Thank you very much for taking the time to review this manuscript. We have included the detailed responses below and the revisions/corrections highlighted/in track changes in the re-submitted files.
  1. Gene names should be written out in full in their first use, followed by abbreviation. Please check the whole manuscript for this point.

Thank you for your kind observation. We have scanned the whole manuscript and added the full names of genes mentioned for the first time, followed by their abbreviation.

  1. Please follow the common use of “lncRNAs” and “circRNAs” instead of “lnc-“ or “circ-RNAs” throughout the manuscript.

Thank you for your observation, we have changed to lncRNAs and circRNAs throughout the review.

  1. Please ensure that the words like “in vitro”, “in vivo”, “de novo” are italicized. 

Thank you for your correction. We have ensured the terms are italicized.

  1. Introduction cover three major concepts, namely cancer, exosomes and ncRNAs. It might be better to use subtitles.

Thank you for your kind suggestion. We have used subtitles in the introduction to cover the concepts that you kindly mentioned.

  1. Although the title covers the clinical relevance of “lncRNAs” and “circRNAs”, the authors describe only lncRNAs in Introduction (lines 75-99). It would be nice to describe “circRNAs” and perhaps touch on the ceRNA hypothesis.

Thank you for your suggestion. We have added some information regarding circRNAs (lines 93-99).

  1. Figure 2: This figure shows the molecular function(s) of lncRNAs not “classification of lncRNAs”. Please either replace the figure or the title.

Thank you for your observation. We have changed the title of Figure 2 to “Molecular functions of lncRNAs”.

  1. Figure 3: The text on the figure is too small to read. Also, I wonder whether the authors have taken permission from the authors of “Ref. 44”.

Thank you for your kind comments. The figures have been designed with the highest possible definition to be observed in detail when zoomed in, including the labels. Additionally, all permissions and authorizations for the use of images have been considered.

  1. Please be consistent in the use of numbers (e.g. separation of 5-digit numbers preferably with a comma—please see the line 188).

Thank you for your observation. We have added a comma in 4- and 5-digit numbers.

  1. Figure 4: Although this is a highly informative figure, it is challenging to track the tissue source of lncRNA/circRNA. Some of the unnecessary components may be omitted to highlight the essential message. Additionally, the resolution should be improved.

Thank you for your comment. We have added small arrows to indicate the lnc- or circRNA related to the specific type of cancer. As aforementioned, the figures have been designed and sent to the editors in the highest resolution possible. The final document may have an HD version of the figures.

  1. Clinical relevance and future perpectives contains only the relevance of lncRNAs without any mention of circRNAs. Please indicate how exosome-cargoed circRNAs can be of relevance.

Thank you for your comment and suggestion. Section 4 mentions several non-coding RNAs used in various studies as potential biomarkers in fluids such as blood and bile, in their association with cancer stages, their potential for invasion, metastasis, and poor prognosis. In addition to the evidence presented, the authors did not identify other lncRNAs or circRNAs identified as biomarkers or therapeutic targets in EVs.

Round 2

Reviewer 3 Report

Comments and Suggestions for Authors

All the points have been properly addressed. I think that the current version warrants publication.

Comments on the Quality of English Language

Minor errors.

Author Response

We thank the reviewer for their comments on our manuscript.